# Dysregulated S100A9 Expression Impairs Matrix Deposition in Chronic Wounds

**DOI:** 10.3390/ijms25189980

**Published:** 2024-09-16

**Authors:** Sandra Franz, Marta Torregrossa, Ulf Anderegg, Anastasia Ertel, Anja Saalbach

**Affiliations:** Department of Dermatology, Venereology and Allergology, Max Bürger Research Centre, Medical Faculty, University Leipzig, Johannisallee 30, 04103 Leipzig, Germany; sandra.franz@medizin.uni-leipzig.de (S.F.); marta.torregrossa@medizin.uni-leipzig.de (M.T.); ulf.anderegg@medizin.uni-leipzig.de (U.A.); anastasia.ertel@medizin.uni-leipzig.de (A.E.)

**Keywords:** chronic wounds, diabetes, iron overload, S100A9, S100A8, extracellular matrix, fibroblasts

## Abstract

Chronic non-healing wounds are characterized by persistent inflammation, excessive matrix-degrading proteolytic activity and compromised extracellular matrix (ECM) synthesis. Previous studies showed that S100A8/A9 are strongly dysregulated in delayed wound healing and impair the proper function of immune cells. Here, we demonstrate an unrecognized pathological function of *S100A9* overexpression in wounds with impaired healing that directly affects ECM functions in fibroblasts. S100A9 was analyzed in two different mouse models mimicking the features of the two most prominent types of non-healing wounds in humans. Db/db mice were used as a model for diabetes-associated impaired wound healing. Iron-overloaded mice were used to mimic the conditions of impaired wound healing in chronic venous leg ulcers. The skin wounds of both mouse models are characterized by delayed wound closure, high and sustained expression of pro-inflammatory mediators and a substantially decreased ECM deposition, all together the hallmarks of non-healing wounds in humans. The wounds of both mouse models also present a solid and prolonged expression of *S100A8* and *S100A9* that coincides with a compromised ECM deposition and that was confirmed in chronic wounds in humans. Mechanistically, we reveal that S100A9 directly affects ECM deposition by shifting the balance of expression of ECM proteins and ECM degrading enzymes in fibroblasts via toll-like-receptor 4-dependent signaling. Consequently, blocking S100A9 during delayed wound healing in db/db mice restores fibroblast ECM functions eliciting increased matrix deposition. Our data indicate that the dysregulation of S100A9 directly contributes to a compromised ECM deposition in chronic wounds and further suggests S100A9 as a promising therapeutic target to improve tissue repair in chronic wounds.

## 1. Introduction

Skin wound repair is a complex, highly dynamic process that enables rapid closure of the skin barrier. It requires spatiotemporal coordination between the cell types involved, including infiltrating immune cells (neutrophils and monocytes/macrophages) as well as skin resident cells, and the extracellular matrix (ECM), cytokines, chemokines, and growth factors [1,2,3]. Wound repair is structured in three sequential, partially overlapping phases: inflammation, tissue formation and remodeling. After injury, platelets are activated and initiate the coagulation cascade leading to the deposition of a provisional fibrin matrix within the wound. As a result, a large number of mediators are secreted to initiate chemotaxis of the neutrophils and monocytes/macrophages, starting the inflammatory phase [4]. Neutrophils are the first cells entering the wound bed. They fulfil different functions, including phagocytosis and the elimination of pathogens as well as the release of cytokines, chemokines and growth factors, before they undergo apoptosis. Apoptotic neutrophils are then cleared by macrophages initiating an inflammation-resolution program [5]. Macrophages differentiate from infiltrating monocytes. While they initially promote inflammation through cytokine release and aid in phagocytosis they later resolve inflammation and produce growth factors that activate pro-repair functions in fibroblasts, angiogenesis and keratinocyte migration [2]. The phase of tissue formation starts about three days post-wounding with immigration and proliferation of wound-associated fibroblasts. Initially, a loose and highly vascularized granulation tissue fills the dermal part of the wound and is subsequently strengthened by the deposition of a rather disorganized provisional ECM composed of high amounts of immature type III collagen secreted by the fibroblasts [6]. Fibroblast differentiation into myofibroblasts enhances collagen deposition and wound contraction. In parallel, the proliferation and migration of keratinocytes towards the wound center enable re-epithelization and wound closure. The final remodeling phase is characterized by the replacement of type III reticular collagen in the granulation tissue by type I fibrillar collagen over the next years, building scar tissue. Matrix-degrading enzymes like matrix metalloproteinases (MMPs) play a central role in this tissue-remodeling process [2,4].

The inability to properly execute the proliferative phase to enable wound closure leads to the formation of non-healing chronic wounds like diabetic foot ulcers (DFU), chronic venous leg ulcers (CVU) and pressure ulcers. In contrast to acute wounds, which commonly heal within three weeks, chronic wounds persist per definition for at least three months or longer. Chronic wounds represent a major socioeconomic burden. They are highly prevalent, primarily affecting the elderly and people with comorbidities such as diabetes, obesity and cardiovascular diseases [1]. Chronic wounds are characterized by persistent inflammation, excessive proteolytic activity from MMPs and compromised ECM deposition [7]. For example, reduced collagen expression and enhanced expression of MMPs have been described at the wound edge of DFUs [8]. Hence, the disturbed balance between ECM production and ECM degradation results in the disorganization of the ECM network in DFUs. Similarly, a proteome analysis of the wound exudates obtained from a human CVU revealed decreased protein levels of different collagens whereas the MMPs were elevated [9]. The prolonged exposure of chronic wounds to pro-inflammatory cytokines, such as IL-1β and TNF-α, contributes to the dysregulation of ECM remodeling in diabetic wounds [10].

S100A8/A9 belong to the family of danger-associated molecular patterns that are induced upon infection, injury or inflammation to initiate the first rapid inflammatory response. S100A8/A9 are secreted by myeloid cells such as neutrophils and monocytes upon infection with bacteria. They stimulate leukocyte recruitment and the secretion of inflammatory cytokines, reactive oxygen species and nitric oxide. Loss of S100A8/A9 regulatory mechanisms results in fatal TNFα-driven inflammation [11,12]. Moreover, the S100A8/A9 complex displays broad-spectrum antimicrobial activity against various microorganisms by binding Zn^2+^ and Mn^2+^ [13,14]. Thorey et al. demonstrate that *S100A8/A9* are strongly expressed during dermal wound healing in the hyperproliferative wound epithelium and in the inflammatory neutrophils and macrophages [15]. S100A8/A9 action is not restricted to the skin. Spinal cord injury is associated with high levels of S100A9. Blocking S100A9 significantly improved motor function, and reduced the cavity formation and neutrophil infiltration in the lesion. A S100A9 blockade upregulated the gene expression of anti-inflammatory genes while inhibiting the expression of pro-inflammatory factors such as *IL-1β*, *IL-6* and *TNF-α-*. S100A9 may be a useful target for the treatment of a spinal cord injury [16]. Similarly, *S100A8* and *S100A9* are the most upregulated genes in the myocardium in the immediate post-ischemic period. Clinical and experimental evidence supports the deleterious role of S100A8/A9 in the immediate post-ischemic period. High levels of S100A8/A9 during the first 24 h of myocardial infarction are associated with increased incidence of major adverse cardiovascular events and heart failure. The mouse models underline the detrimental role of S100A9 in the first phase of myocardial infarction [17].

In a previous study, we demonstrated the strong dysregulation of S100A8 and A9 in imiquimod-induced skin inflammation in obese mice [18]. In both wild-type (WT) and obese mice, *S100A8* and *S100A9* expression are rapidly induced upon challenge but showed significantly higher levels of gene and protein expression in the lesional skin of obese mice compared to WT mice. Furthermore, while expression of S100A9 declines in the later phase of inflammation in wild-type mice it persists in obese mice until late time points [18].

We showed that the overexpression of *S100A8/A9* during inflammatory conditions in the skin of mice with obesity impairs appropriate macrophage activation and the polarization required for inflammatory resolution and tissue repair and contributes to impaired wound healing in type 2 diabetic mice [18]. Overexpression of *S100A8* and *A9* in several inflammatory diseases such as psoriasis, rheumatoid arthritis, osteoarthritis, obesity and Alzheimer’s disease reflects the pathological function of S100A8/9 and indicates that targeting these proteins may open new therapeutic options in inflammation-associated diseases [14,15,19].

In the present study, we observe a prolonged and increased expression of S100A8 and A9 in impaired wound healing in db/db mice and iron-overloaded mice, resembling wound healing in DFUs and CVUs, respectively. We, therefore, explored the direct action of S100A9 on ECM deposition in delayed wound healing, apart from its action on macrophages shown before [18].

## 2. Results

### 2.1. Production and Deposition of ECM Is Attenuated in Impaired Wound Healing

Wound healing was analyzed in two different mouse models resembling features of disturbed wound healing in DFUs and CVUs.

Db/db mice represent an established model of type 2 diabetes (T2D)-associated impaired wound healing. Wound closure is attenuated in the db/db mice (Figure 1A,B). Trichrome staining of the tissue sections demonstrates the strongly reduced collagen deposition 10 days after wounding the db/db mice compared to the wild-type animals (WT, Figure 1C). Consistent with the histological data, gene expression of collagen I and III was decreased in the db/db mice 10 days post-wounding (Figure 1D).

Chronic venous leg ulcers (CVUs) are characterized by a pathological iron overload in the skin tissue. To mimic the conditions of iron overload in the skin and to investigate the underlying pathological mechanism of disturbed tissue formation in CVUs, a previously established model of systemic iron overload was used; wherein, mice were subjected to repeated treatment with iron-dextran (FeDx), which resulted in an abundant accumulation of iron in the skin [20]. The skin of the systemic iron-overloaded mice presented high levels of iron deposition similar to the skin of patients with CVUs (Appendix A). Full-thickness wounding of the mice was performed, and wounds were analyzed five- and seven-days post-wounding. As shown in Figure 1E/F wound closure in the iron overloaded skin is delayed. Furthermore, the deposition of collagen and gene expression of collagen I and III is markedly reduced in the iron-overloaded wounds (Figure 1G, H).

Altogether, wounds of both mouse models, diabetic db/db mice and iron-overloaded mice are characterized by compromised wound closure and attenuated matrix deposition, as observed in the non-healing wounds in diabetic patients and patients with CVUs.

### 2.2. Overexpression of S100A8 and S100A9 during Impaired Wound Healing Is Associated with Diminished ECM Production

To understand the underlying mechanisms of dysregulated ECM deposition in non-healing wounds, we first compared the gene expression pattern of wounds 10 days after wounding in WT and diabetic db/db mice. A total of 1044 genes were at least threefold regulated, 606 were upregulated in the wounds from the db/db mice compared to the WT mice, while 438 were downregulated (Figure 2A, Appendix A). *S100A8* and *A9* are found among the top ten upregulated genes in the wounds from the db/db mice (Figure 2B). Immunofluorescence staining of S100A9 in the unwounded and wounded skin of the WT mice shows that S100A9 is not present in the skin at a steady state but is induced upon injury (Figure 2C). S100A9 protein expression strongly increases in the wounds of the WT mice until day five and decreases afterwards. On day 10, post-wounding, S100A9 almost completely disappeared. In contrast to the WT mice, wounds of the db/db mice on day 10 showed strong S100A9 signals in the immunofluorescence analysis. S100A9 was found in the dermal and epidermal compartments of the diabetic wounds (Figure 2C). A gene expression analysis supported the results from the immunofluorescence staining. Confirming previous data, S100A8 and A9 are strongly induced in WT mice after injury and their expression declines to basal level 10 days after wounding. In contrast, in the wounds from the db/db mice, gene expression of S100A8 and S100A9 is significantly higher compared to the WT wounds and does not decline, resulting in a persistently high S100A8 and A9 expression even in the later stages of wound healing (Figure 2D [18]).

To investigate the underlying mechanism of high S100A8 and A9 expression in diabetic wounds, we used ex vivo skin cultures that we stimulated with typical diabetes-associated inflammatory and metabolic factors. This showed that TNFa in combination with IL-1β induces high S100A9 expression in the epidermis of the skin while metabolic factors like insulin and high glucose did not affect S100A9 expression (Figure 2E). Consistently, we found a high and prolonged expression of TNF-α and IL-1β in the wounds from the db/db mice compared to the WT mice (Figure 2F). Thus, the pro-inflammatory microenvironment in diabetic wound healing contributes to the high and prolonged S100A9 expression.

We next asked whether S100A8 and S100A9 are also increased in wounds from iron-overloaded mice, mimicking features of human CVUs. Indeed, wounds seven days post-wounding showed a clearly increased expression of S100A8 (significantly) and S100A9 (by trend) in the iron overloaded mice in comparison to the control mice (Figure 3A). To confirm the relevance for human CVUs, we analyzed S100A9 protein expression in healthy skin, acute wounds and CVUs via immunofluorescent staining of the histological sections (Figure 3B). S100A9 is hardly detectable in healthy skin. In acute wounds, S100A9 is clearly induced and can be detected primarily in a cellular association in the dermal compartment and partly in the epidermis. In contrast, S100A9 appears highly elevated in CVUs and is predominantly present in the hyperproliferative epidermis of the wound margins and with few positive cells in the dermis (Figure 3B).

Since iron is one of the main pathological drivers in CVUs [20] we investigated the role of iron overload in the epidermal induction of S100A8 and A9. The stimulation of murine ex vivo skin cultures with high doses of iron resulted in the significant upregulation of S100A8 and S100A9 gene expression in the epidermis of the skin, confirming iron as a direct cause of epidermal S100A8 and A9 overexpression (Figure 3C). However, in the diabetic db/db mouse model we showed that epidermal S100A8/A9 were mainly induced by pro-inflammatory factors in the wound environment. Increased inflammation in the wound environment of CVUs associated with the unrestrained pro-inflammatory activation of macrophages has been reported [20]. We therefore asked whether the iron overload might also contribute to an increase in the pro-inflammatory signals via modulating macrophage activation and thus upregulates *S100A8* and *S100A9* expression through this indirect mechanism. We isolated bone marrow cells from the WT mice to generate wound-healing associated macrophages via differentiation with M-CSF and IL-4. The stimulation with high concentrations of iron induced the expression of TNF but not of IL-1β in these macrophages (Figure 3D). In line with this, we detected increased gene expression of TNF but not of IL-1β in the wounds of the iron-overloaded mice on day seven post-wounding (Figure 3E). However, stimulation of the murine ex vivo skin cultures with TNF induced only a slight upregulation of S100A8 and S100A9 gene expression in the epidermis of the skin (Appendix A).

In summary, iron-overload directly induces epidermal overexpression of S100A8 and S100A9 while iron-induced TNFa from macrophages seems to play a rather marginal role in S100A8/A9 induction in iron-overload conditions such as in CVUs.

Taken together, both diabetic wounds in the db/db mice and wounds in the mice with iron overload present overexpression of *S100A8* and *S100A9* while expression of collagen and ECM production are reduced (Figure 1D, Figure 2D and Appendix A). Importantly, in the wounds of the corresponding control mice of both mouse models, the decline of *S100A8* and *A9* expression coincides with the increase of ECM protein expression, suggesting the possible role of S100A8 and A9 in impaired tissue formation in diabetic and iron-overloaded mice (Figure 1D, Figure 2D and Appendix A).

### 2.3. S100A9 Directly Affects the Deposition of ECM by Dermal Fibroblasts during Impaired Wound Healing

Fibroblasts are the main regulators of ECM deposition through the balanced expression of ECM proteins and matrix-degrading enzymes. To investigate whether these fibroblast functions are dysbalanced when tissue formation is impaired, we isolated fibroblasts from the wounds of db/db and wild-type mice, removing unwanted cells using a negative selection protocol. The purity of the fibroblast population was confirmed by a PCR analysis that showed a high expression of the typical fibroblast markers *Thy1* and *Collagen I* in the fibroblast fraction, while keratinocyte, endothelial cell and leukocyte markers were barely detectable (Appendix A). Consistent with the histological data in Figure 1A showing reduced ECM deposition in db/db wounds, expression of ECM proteins and matrix anabolic enzymes like Plod2 is significantly decreased in the fibroblasts from the db/db mice (Figure 4A). In parallel, gene expression of the collagen degrading enzymes Mmp10 and Mmp13 are increased in the fibroblasts isolated from the wounds of the db/db mice (Figure 4A).

To explore whether S100A8 and S100A9 contribute to this dysbalanced fibroblast activation, murine dermal fibroblasts were treated with S100A8, S100A9 and S100A8/9 dimer (Figure 4B). S100A9 significantly inhibited the expression of *Col3a1* and *Plod2.* In addition, S100A9 strongly stimulated the expression of matrix-degrading enzymes such as *Mmp3*, *Mmp9*, *Mmp10* and *Mmp13*. In contrast, S100A8 and the S100A8/A9 dimer affected neither the expression of the ECM protein nor the expression of MMPs, pointing to the important role of S100A9 in the dysregulation of fibroblast activation. To unravel the contribution of TLR4 (toll-like receptor 4) and RAGE (receptor for advanced glycation end products) as important recognition receptors of S100A9, fibroblasts were stimulated with S100A9 in the presence of the TLR4 inhibitor CLI-095 and RAGE antagonistic peptide. The blocking of TLR4 completely abolishes the S100A9-induced upregulation of MMPs expression and the downregulation of *Plod2* expression, while inhibition of RAGE only partially blocks MMP13 induction. Only the effects on *Col3a1* induced by S100A9 were not affected by either TLR4 or RAGE inhibition. In summary, this shows that S100A9 mediates its effect on fibroblasts predominantly via TLR4 signaling. Interestingly, dermal fibroblasts from the diabetic db/db mice displayed enhanced TLR4 expression (Appendix A). Moreover, the blocking of S100A9 by paquinimod reduces TLR4 expression in the fibroblasts (Appendix A). However, this effect seems to be indirect since S100A9 does not stimulate TLR4 expression (Appendix A).

Finally, we proved the link between impaired ECM deposition and pathological S100A9 overexpression in the compromised healing of the wounds in the db/db model. Full-thickness excisional wounds were inflicted on the back of the db/db mice. The mice were randomly divided into a treatment group and a control group. In the treatment group S100A9, activity was blocked by the administration of paquinimod via drinking water. Paquinimod belongs to a group of Quinoline-3-carboxamides (Q-compounds). It prevents the binding of S100A9 to its receptors, TLR4 and RAGE [21]. To exclude the effects caused by the chemotactic and pro-inflammatory activity of S100A9, paquinimod treatment was started two days after wounding when an initial inflammatory response was already established. Blocking S100A9 increases *Col3a1* and *Plod2* expression in the fibroblasts of the paquinimod-treated db/db mice versus the non-treated db/db mice (Figure 4C). In addition, the expression of matrix-degrading enzymes such as *Mmp10* and *Mmp13* is significantly reduced in the wound fibroblasts of the treatment group (Figure 4C).

Thus, S100A9 supports matrix degradation through the stimulation of matrix-degrading enzymes. In parallel, it inhibits the expression of matrix proteins and its modifying enzymes. In sum, the direct action of S100A9 on fibroblasts attenuates ECM deposition.

## 3. Discussion

The regulation of matrix deposition is a key event in many physiological and pathological situations. It is required for normal wound healing where the ECM components are produced during the formation of granulation tissue and the final replacement of the provisional matrix by mature connective tissue. In early granulation tissue, the newly deposited ECM is predominantly composed of proteins such as fibronectin and vitronectin as well as collagens IV, V and VI, forming scar tissue. In the later phases of wound healing, the scar tissue is remodeled, and the ECM proteins are gradually replaced by collagen I. Diminished ECM deposition results in non-healing wounds and skin atrophy while excessive deposition of connective tissue is the pathological hallmark of fibrosis, hypertrophic scars and keloids. A tight balance between connective tissue synthesis and degradation is therefore required for the normal functioning of all tissues [22].

Chronic wounds such as venous leg ulcers, diabetic foot ulcers and pressure ulcers do not proceed through the phases of a normal wound-healing process but remain in an uncontrolled inflammatory stage [23]. In addition to persistent inflammation, the various chronic wounds of different etiologies share further common features, such as deficient local angiogenesis and compromised new tissue formation; both are typically driven by the inflammatory response. The high serum levels of TNFα, MCP-1, MMP-9 and FGF-2 were found to be associated with a poor outcome of healing of diabetic foot wounds [24]. Pro-inflammatory mediators in diabetic wounds increase the expression of matrix-degrading enzymes resulting in decreased collagen deposition [7,9]. Similarly, CVUs are characterized by persistent inflammation, large numbers of senescent cells and excessive proteolytic activity from MMPs that all together impair tissue formation [10]. Hence, the highly proteolytic environment in chronic wounds with elevated levels of MMPs together with decreased levels of tissue inhibitors of metalloproteinase (TIMPs) shift the balance between ECM deposition and degradation towards excessive ECM breakdown resulting in attenuated wound healing in both diabetic wounds and CVUs [25]. In the present study, we used type 2 diabetic db/db mice to mimic diabetes-associated wound healing and mice with systemic iron overload, including the skin, to imitate disturbed wound healing in chronic venous leg ulcers. Both models are characterized by delayed wound closure, high and sustained expression of pro-inflammatory mediators such as *TNFα* and *IL-1β*, and a substantially decreased ECM deposition; all of which resemble features found in non-healing wounds in humans.

In a previous study, investigating mechanisms of increased skin inflammation in mouse models of obesity including db/db mice, we described an increased and prolonged expression of the alarmins, *S100A8* and *S100A9*, during diabetic wound healing in db/db mice [18]. S100A8 and A9 belong to the family of danger associated molecular patterns that are rapidly induced upon infection, injury or inflammation initiating the first rapid inflammatory response [14]. Thorey et al. describe a strong upregulation of the RNA and protein levels of S100A8 and S100A9 in acute mice and human wounds [15]. In line with this study, we also find a rapid and strong induction of S100A8 and S100A9 expression in the acute skin wounds of the WT mice after injury, which declines within seven days. In contrast, in delayed wound healing in the db/db mice, expression of S100A8 and S100A9 is significantly higher and remains high even at the late stages of wound healing. Consistently, S100A8 and S100A9 belong to the most upregulated genes in the comparative gene sequencing analysis between db/db mice and WT mice. Krisp et al. found through a proteome analysis a tenfold increase of S100A8 and A9 in the chronic wounds of diabetic patients compared to acute wounds, which supports the importance and relevance of our results in the mouse model in regards to the situation in human patients [26]. Mechanistically, we showed in the previous study, that *S100A8* and *A9* overexpression is induced via a self-perpetuating circle involving S100A8 and A9, free fatty acids and IL-1β. In brief, S100A8/A9 released upon injury or inflammation stimulates IL-1β gene expression in macrophages via the TLR4-NFkB pathway. The presence of high levels of saturated free fatty acids in obese mice activates the inflammasome in the macrophages resulting in the release of IL-1β. Macrophage-derived IL-1β stimulates and amplifies S100A9 expression via the NFkB and MAPK signaling pathways in the tissue, thereby initiating a sustained S100A9 response during the inflammatory responses in the skin under obese conditions. In line with these observations, the db/db mice present an increased expression of IL-1β and TNF during delayed wound healing. Further, stimulation of skin cultures with TNF and IL-1β, but not with diabetes-associated factors like insulin and glucose, stimulate *S100A8* and *A9* expression in the skin, supporting the previous notion of an inflammatory-driven pathological overexpression of S100A8 and A9.

Less is known about the role of S100A9 in the pathogenesis of chronic venous leg ulcers. Mimicking the pathophysiological environment of CVUs, the wounds of iron-overloaded mice revealed a high expression of *S100A8* and *A9* in comparison to the control wounds in the WT mice. Consistently, tissue sections of CVUs from patients display a high expression of S100A9 in the hyperproliferative wound epithelium and in dermal cells, supporting the data from the mice. We further identify the high iron levels in CVUs as one of the main causes of the overexpression of *S100A8* and *A9*. The ex-vivo skin cultures indicate that iron overload directly induces epidermal *S100A9* expression. Furthermore, high iron concentrations reshape the function of IL-4-induced pro-repair macrophages by inducing the expression of *TNF*-α. Consistent with our data, a previous study demonstrated iron-induced unrestrained pro-inflammatory macrophage activation with elevated TNF-α in wounds from iron-overloaded mice and in humans with CVU [20]. However, TNF-α seems to play a marginal role in the control of *S100A8/A9* expression since TNF-α only weakly induced *S100A8/A9* expression in the ex vivo skin cultures.

S100A8 and A9 are well-described for their pro-inflammatory and chemotactic action on immune cells [13,27]. In our previous study, we showed that *S100A9* overexpression impairs the differentiation of inflammation-resolving and tissue repair-promoting M2-like macrophages in favor of promoting inflammatory functions in these macrophages [18]. This suggests that impaired tissue formation in wounds with high S100A8 and A9 might result from increased inflammation and disturbed macrophage activation that regulate the activation of tissue cells to produce new ECM. However, here, we describe also a direct effect of S100A9 on matrix deposition during dermal wound healing independent of its effect on inflammatory cells through the regulation of tissue cells. We reveal a mechanism to address how the increased activity of the inflammatory factor S100A9 directly compromises the wound healing functions of the resident skin fibroblasts, and how blocking this factor can restore the activity of these cells. The role of S100A8 and A9 during wound healing and the consequences of the dysregulation of S100A8 and A9 expression on tissue cell functions are still poorly understood. There is little data about the effect of S100A8 and A9 on fibroblasts and consequently on the control of ECM production and degradation, the final step for adequate tissue repair. In the wounds of normal healing WT mice, ECM gene expression starts at the same time as the expression of S100A8 and A9 declines. The prolonged expression of S100A8 and A9 in the wounds of the diabetic db/db mice and of the iron-overloaded mice suggests a relation between ECM production and the dysregulation of *S100A8* and *A9* expression. The fibroblasts isolated from the diabetic wounds presented a diminished expression of the ECM-forming proteins collagen and Plod2 in parallel to an increased expression of ECM degrading MMP, indicating a defective ECM deposition under diabetic conditions. The inhibition of S100A9 activity with paquinimod, which blocks the major S100A9 receptors TLR4 and RAGE [21], restored the expression of ECM proteins and matrix-degrading enzymes in the wound fibroblasts in the db/db mice, proving the role of S100A9 in the control of ECM deposition. Paquinimod was applied two days after wounding, when the initial inflammatory phase was already established, to exclude the blocking effects of S100A9 on the initiation of inflammation and the activation of immune cells.

Mechanistically, we demonstrate a strong induction of MMPs expression together with a strong downregulation of collagen III and Plod2 expression by S100A9 in dermal fibroblasts in vitro. The increased expression of MMPs mediated by S100A9 is supported by previous studies. For example, S100A9 has been shown to enhance *MMP9* and *MMP3* expression in fibroblast-like synoviocytes and tendocytes, respectively [28,29]. Consequently, blocking S100A9 by paquinimod reduces the expression of matrix-degrading enzymes in the fibroblasts isolated from the diabetic wounds. TLR4 and RAGE have been described as receptors for S100A9 [14,30]. The inhibition experiments indicate that S100A9-mediated induction of MMPs in skin fibroblasts depends mainly on TLR4 signaling and partially on RAGE signaling.

Moreover, blocking S100A9 during wound healing in the db/db mice increased gene expression of collagen III and Plod2. Collagen III is the main collagen in early granulation tissue which is replaced in later stages by collagen I. Plod2 catalyzes lysyl hydroxylation of procollagen, resulting in the formation of stabilized collagen cross-links and enhancing the stiffness of the matrix [31]. There are opposite results on the effects of S100A9 on ECM production. Xu et al. describe an increase of collagen III in lung fibroblasts caused by S100A9 treatment [32]. Injection of S100A9 in mechanically stressed rat skin rescued dermal thinning and collagen expression [32,33]. In contrast, the treatment of tendocytes with S100A9 did not affect collagen I and III expression [29]. This indicates that the action of S100A9 on fibroblasts also depends on the microenvironment. For example, S100A9 exhibits the affinity for several divalent metal ions, including Ca^2+^, Mn^2+^, Zn^2+^ and Cu^2+^, which might affect its stability, structure and fibril formation ability [34]. Elevated calcium limits the potential for S100A9 to adopt a fibrillar structure. Whether changes of these metal ions occur in chronic wounds, and thus might affect S100A9 structure and function, is still an open question.

Interestingly, skin wound healing is significantly delayed in the TLR4-deficient mice, with decreased infiltration of the neutrophils and macrophages most likely due to the disturbed initial inflammatory response that is required for normal wound healing. In contrast, TLR4 deficiency and pharmacological inhibition of TLR4 improve disturbed wound healing in high-fat diet-induced obese mice with insulin resistance [35,36]. TLR4 expression and signaling are pathologically increased in diabetic patients and db/db mice [36]. We observed an increased expression of TLR4 in the fibroblasts isolated from the wounds of db/db mice making them more susceptible to S100A9 action. Taken together, overexpression of S100A9 and its receptor, TLR4, might contribute to S100A9-mediated attenuated ECM deposition in non-healing wounds. However, the mechanisms of TLR4 overexpression are still unknown. S100A9 does not affect TLR4 expression in fibroblasts.

S100A8 and S100A9 are found as homodimers but preferentially form the S100A8/A9 heterodimer (also called calprotectin) in the presence of divalent cations [18]. In the present study, we show that that only S100A9 but not the S100A8 homodimer or S100A8/A9 heterodimer are sufficient to alter ECM and MMP expression in dermal fibroblasts. Importantly, several studies describe the identification of the S100A9 homodimer, S100A9 monomers, and S100A8/S100A9 heterodimers in different in vivo situations [37], suggesting that all forms might be functionally relevant. Indeed, the isolated action of the S100A9 monomer [21], S100A8 monomer [11] or both [29] has been described. The action of S100A8 and A9 might be dependent on the cell type, tissue and microenvironment. This plasticity might contribute to the diverse and sometimes opposite functions described for S100A8/A9 [38].

S100A9 is already known for its roles as a chemoattractant, pro-inflammatory mediator, and regulator of macrophage differentiation. In our study, we demonstrate a further pathological function of S100A9 overexpression in impaired healing by affecting ECM deposition in fibroblasts. S100A9 directly acts on the fibroblasts via TLR-4 signaling, shifting the balance of expression of the ECM proteins and ECM degrading enzymes and resulting in a reduced ECM deposition and subsequently hindered wound healing. Lately, fibroblasts have been increasingly recognized as important regulators of inflammation and repair processes, and understanding their dysregulation in impaired wound healing potentially opens up new targets for therapies [39]. Importantly, blocking S100A9 can restore fibroblast ECM function and reverse the negative effects on wound healing. This suggests S100A9 is a promising therapeutic target to improve disturbed tissue repair in diabetic wound healing and in CVUs.

In our study, we described the local overexpression and uncontrolled activity of S100A9 contributing to the disturbed healing of chronic wounds. Consequently, local inhibition of S100A9 activity in the skin wounds would be a favorable approach for clinical transition. Paquinimod, Tasquinimod and ABR-238901 are potent S100A9 inhibitors that block the interaction of S100A8/A9 with their receptors RAGE and TLR4. Treatments with the S100A8/A9 blockers have demonstrated encouraging results in experimental and clinical interventional studies on cardiovascular diseases, systemic inflammatory, autoimmune diseases and cancer [40]. For example, paquinimod has been effectively tested in phase II clinical trials for the treatment of systemic sclerosis [41]. Therefore, paquinimod may also represent a promising treatment option for chronic wounds with increased S100A9 activity as we showed in our diabetic wound healing model in the mouse. However, Paquinimod, Tasquinimod and ABR-238901 are orally active substances and act systemically, bringing the risk of unwanted systemic side effects. One important challenge in using S100A8/A9 as a therapeutic target in chronic wounds is the relative abundance of this protein in human circulation, with median levels of approximately 5 mg/L in healthy individuals and up to 15 mg/L in atherosclerosis patients [40]. From this perspective, the development of topical application methods for S100A9 inhibitors for local application to skin or wounds would be an important approach for the clinical implementation of our findings.

## 4. Materials and Methods

### 4.1. Human Studies

Skin specimens from fresh, clinically health skin, or from one- to three-day-old acute wounds were collected from patients who underwent flap surgery at the Department of Dermatology, Venereology and Allergology at the University Hospital of Leipzig University. Only remnant skin, not required for diagnostic purposes, was used for the analysis. Skin specimens of chronic wounds were obtained from patients diagnosed with non-healing venous leg ulcers that, despite conventional therapy, persisted for more than 4 months. Remaining tissue samples from biopsies taken from the edge of the ulcer for diagnostic purposes were used for the analysis. All human specimens were collected after written informed consent by the patients was received and it was approved by the Ethics Committee of the University of Leipzig (26 May 2020, ek092-20) according to the Declaration of Helsinki Principles.

### 4.2. Mouse Studies

To generate the mouse model with systemic iron overload, 6-8-week-old C57BL6/J mice were injected intraperitoneally with 5 mg iron-dextran (FeDx) at 200 μL/mouse (Sigma-Aldrich Chemie GmbH, Taufkirchen, Germany USA) every 3 days for 21 days. Wounding of mice was performed as described [18]. All animal experiments were performed according to institutional and state guidelines and were approved by the Committee on Animal Welfare of Saxony (Leipzig, Germany, TVV33/17, TVV13/19, T05/20).

### 4.3. Isolation of Skin Cells from Wounds

Skin wounds were enzymatically digested by 0.15 mg/mL liberase (Roche, Mannheim, Germany) and 0.12 mg/mL DNase (Sigma-Aldrich Chemie GmbH, Taufkirchen, Germany) for 2 h [42]. Fibroblasts were isolated by negative selection. Myeloid cells were removed by magnetic cell separation using a CD11b+ Cell Isolation Kit (Miltenyi, Bergisch Gladbach, Germany) according to the manufacturer’s instructions. PCR analysis confirmed the purity of the cells (Appendix A). Endothelial cell marker (*Pecam*), keratinocyte marker (*Krt5*), and marker for inflammatory cells (*Itgam, Ptprc*) were only barely detectable. Fibroblast markers such as *Thy1* and *Col1a1* were highly expressed in the CD11b− cell fraction, proving the strong enrichment of fibroblasts.

### 4.4. Cell Culture

To isolate skin fibroblast, the skin of WT mice was incubated with 0.15 mg/mL Liberase (Roche, Diagnostics, Mannheim, Germany) for 2 h. Cell suspension was passed through a 70 mm filter to remove tissue debris. Cells were cultured at 37 °C, 5% CO_2_ in DMEM medium (Anprotech, Bruckberg, Germany) containing 10% FCS (Anprotec, Bruckberg, Germany) and 1% penicillin/streptomycin (Biochrom, Berlin, Germany). After reaching confluence, cells were passaged using 0.05% trypsin and 0.02% EDTA (Pan-Biotech, Aidenbach, DE). Fibroblasts were stimulated with 1.5 µg/mL S100A8, S100A9 or S100A8/9 (R&D System, Wiesbaden-Nordenstadt, DE) in the presence or absence of TLR4 signaling inhibitor (CLI-095, 1 µM, Invivogen, Toulouse, France) or RAGE antagonistic peptide (ELKVLMEKEL, 10 µM, R&D System, Wiesbaden-Nordenstadt, DE), for 24 h. In addition, fibroblasts were stimulated with 10ng/mL TNFa and IL-1β (Miltenyi, Bergisch Gladbach, Germany).

Macrophages were generated from the bone marrow of WT mice as described [43]. To differentiate wound-healing associated macrophages, cells were stimulated with 20 ng/mL murine IL-4 (Miltenyi, Bergisch Gladbach, Germany) for 24 h. Macrophages were then stimulated with 50 µM FeSO_4_ or 100 µM FeSO_4_ for 24 h at 37 °C, 5% CO_2_.

### 4.5. Ex-Vivo Skin Culture

Ex-vivo skin culture was performed as described [18]. Mouse skin was stimulated with 10 ng/mL TNFα or 10 ng/mL IL-1β together with 10 ng/mL TNFα (TI), 1 µg/mL insulin, 25 mM glucose, 50 µM FeSO_4_ or 100 µM FeSO_4_ for 24 h at 37 °C, 5% CO_2_.

### 4.6. RNA Preparation and Quantitative Real-Time PCR

RNA isolation from homogenized wounds, epidermis and cells, cDNA generation and quantitative real-time PCR were performed as described [18]. LunaScript RT Supermix (NEB, Frankfurt am Main, Germany) was used to generate cDNA from 500 ng RNA according to the manufacturer′s protocol. Quantitative real-time PCR was performed on a real-time thermocycler using qPCR-soft software with LunaUniversal qPCR Mastermix (NEB, Frankfurt am Main) according to the manufacturer′s instructions. Primer sequences are provided in Appendix A. All PCR products are intron-spanning. Quantitative gene expression was calculated from the standard curve of cloned cDNA and normalized to the reference gene RS36. Data calculation was performed in Microsoft Excel (2016, Microsoft, Redmond, WA, USA) and transferred to GraphPad Prism 10 (Dotmatics, Boston, MA, USA) for graphical representation.

### 4.7. Genome-Wide Expression Analysis

The genome-wide expression analysis was carried out as described by Core Unit DNA Technologies (Faculty of Medicine; University of Leipzig [18]). For data analysis, Affymetrix Gene Chip data were extracted from fluorescence intensities and were scaled to normalize data for inter-array comparison using Transcriptome Analysis Console (TAC) 4.0.2 software according to manufacturer’s instruction (Thermo Fisher Scientific, Darmstadt, Germany).

### 4.8. Tissue Staining

Staining of tissue sections with Masson trichrome and S100A9 was performed as described [18]. Images were captured using the KEYENCE BZ-9000 fluorescence microscope (Keyence GmbH, Leipzig, Germany) and processed using the system software.

### 4.9. Statistics

Statistical analysis for two-group comparisons regarding normally distributed metrical data was performed using a two-tailed Student’s *t*-test. Normality was tested by D’Agostino & Pearson Normality test or Shapiro–Wilk Test (*n* ≤ 4). Where normality was absent, the Mann–Whitney test was used. For statistical comparison of more than two groups, ANOVA Test was used. Calculations were performed using GraphPad Prism version 10. *p*-values of 0.05 or smaller were considered statistically significant. The different degrees of significance were indicated as follows: * *p* < 0.05; ** *p* < 0.01; *** *p* < 0.001, **** *p* < 0.0001.

## Figures and Tables

**Figure 1 ijms-25-09980-f001:**
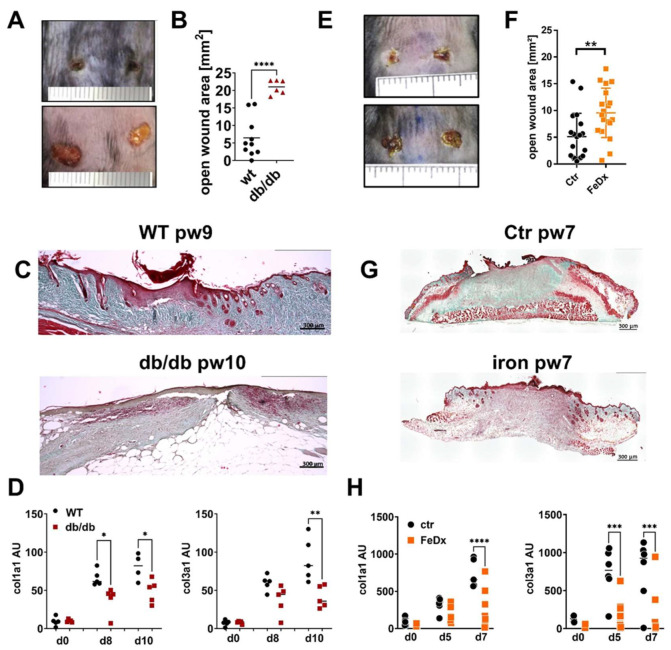
ECM deposition is compromised in mouse models with delayed wound healing. (**A**–**D**) Wild-type (WT) and db/db mice were wounded with 6mm punch biopsies. Wounds were analyzed 10 days post-wounding (pw). (**A**) Wounds of WT and db/db mice. One representative example of five is shown. (**B**) Open wound area 10 days after wounding. (**C**) Masson’s trichrome staining of tissue sections of wounds. One representative example out of 5 WT and 5 db/db is shown. (**D**) Relative expression of indicated genes and time points detected by quantitative PCR in wounds of WT and db/db mice. (**E**–**H**) Control mice (Ctr) and mice systemically treated with iron-dextran (FeDx) were wounded using 6mm punch biopsies. Wounds were analyzed after indicated time points. (**E**) Wounds of control and FeDx-treated mice 7 days after wounding. One representative example of five is shown. (**F**) Open wound area 7 days after wounding. (**G**) Masson’s trichrome staining of tissue sections of wounds on day 7 after wounding. One representative example out of five control (ctr) and five FeDx-treated mice is shown. (**H**) Relative expression of indicated genes detected by quantitative PCR in wounds of control and FeDx-treated mice. Each dot represents one mouse. B/F: Unpaired *t*-test * D/H: ANOVA test with multiple comparisons was performed. *p* < 0.05, ** *p* < 0.01, *** *p* < 0.001, **** *p* < 0.0001. C/D/G/H: Scale bar = 300 µm.

**Figure 2 ijms-25-09980-f002:**
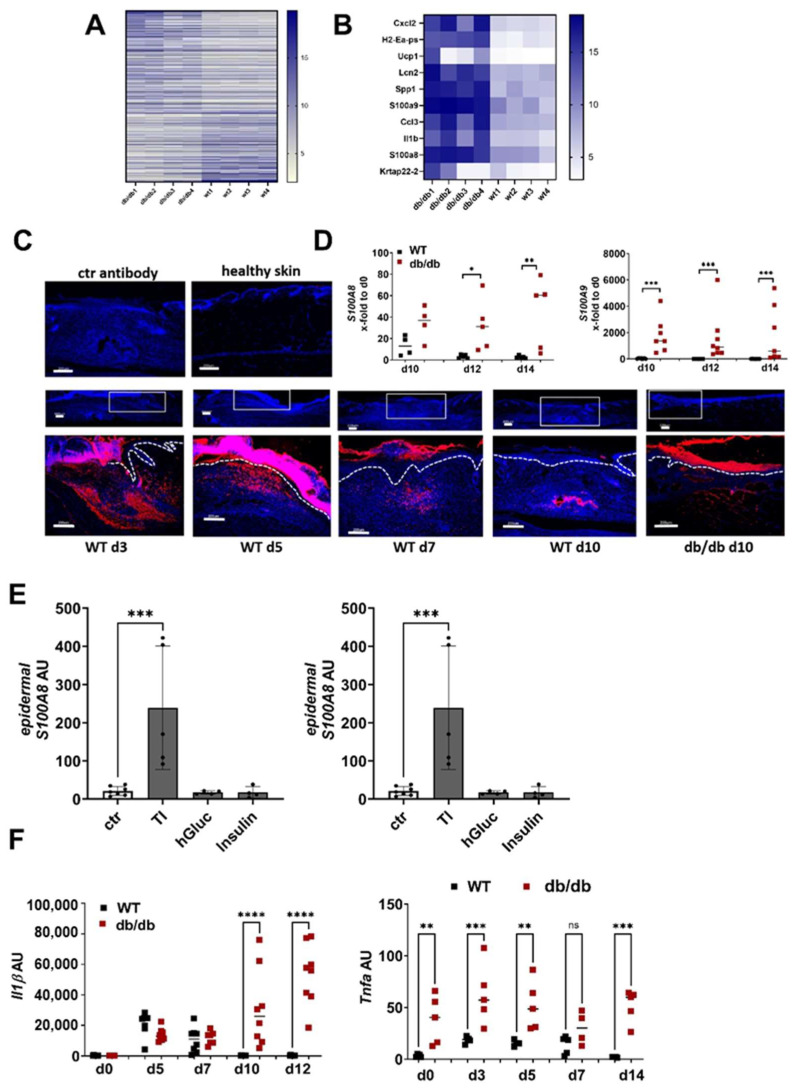
*S100A8* and *A9* are highly expressed in diabetic wounds (**A**–**D**) Wild-type (WT) and db/db mice were wounded with 6mm punch biopsies. Healthy skin and wounds 10 days post-wounding were analyzed. (**A**,**B**) Gene expression analysis by microarray. Differentially expressed genes with 3-fold change were log2-transformed and subjected to heat map analysis (**A**). The 10 most upregulated genes in db/db mice vs. WT mice are shown (**B**). (**C**) Immunofluorescent staining of S100A9 (red) in unwounded skin and the wounds of WT and db/db mice were analyzed after indicated time points. Isotype control antibody was used as the negative control. The box indicates the region within the wounds shown in the lower image. The white dotted line indicates the border between the epidermis and the dermis. Nuclei were stained with DAPI (blue). One representative example of 5 WT and 5 db/db mice is shown. Scale bar = 200 µm. (**D**) Relative expression of indicated genes detected by quantitative RT-qPCR in wounds of WT and db/db mice and X-fold expression at indicated time points compared to d0. (**E**) Mouse skin was cultured ex vivo with TNF-α/IL-1β (TI), high glucose (hGluc) or insulin for 24 h. *S100A8* and *A9* gene expression in the epidermis was analyzed by RT-qPCR. Mean ± standard deviation is shown. ANOVA with multiple comparisons was performed. (**F**) Relative expression of indicated genes detected by quantitative RT-qPCR in wounds of WT and db/db mice. (**D**,**F**) Each represents one mouse. Unpaired *t*-test for each time point was performed. * *p* < 0.05, ** *p* < 0.01,*** *p* < 0.001, **** *p* < 0.0001.

**Figure 3 ijms-25-09980-f003:**
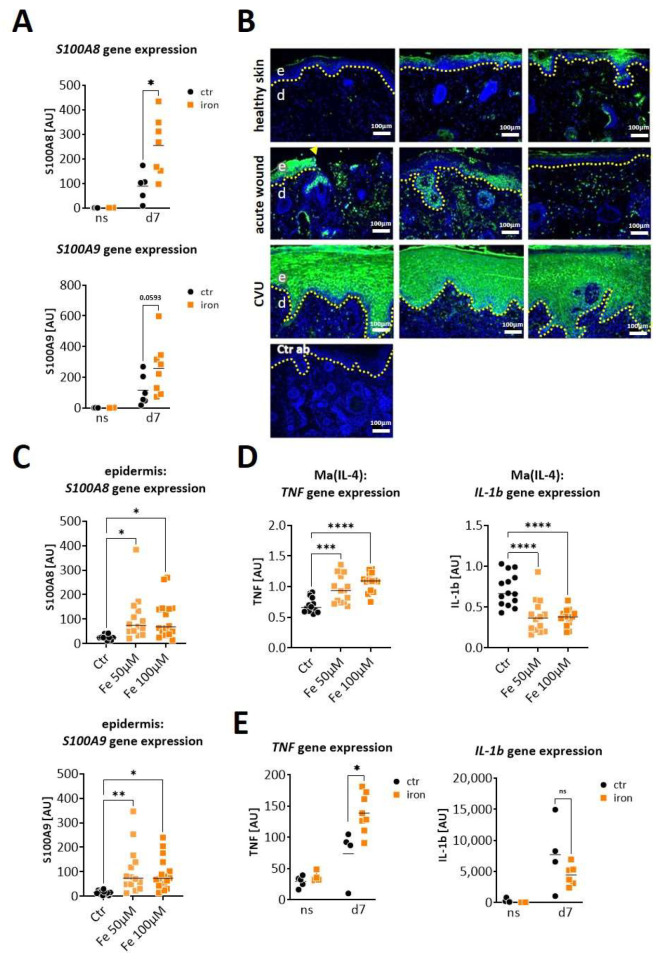
*S100A8* and *S100A9* are overexpressed in wounds in conditions of iron overload. (**A**,**E**) Mice with systemic iron overload and control mice were wounded with 6mm punch biopsies. Normal skin (ns) and wounds were analyzed 7 days post-wounding. (**A**) Relative gene expression of S100A8 and S100A9 was detected by quantitative RT-qPCR in wounds of control (Ctr) and iron-mice. Each dot represents one wound. Mann–Whitney test: * *p* < 0.05. (**B**) Immunofluorescent staining of S100A9 (green) in tissue section from unwounded healthy skin, from human acute wounds and from human chronic venous ulcer. Three representative and independent samples for each condition are shown. Isotype control antibody (Ctr ab) was used as the negative control. Nuclei were stained with DAPI (blue). The dotted line marks the border between the epidermis and the dermis. The arrowhead marks the wound edge, e = epidermis and d = dermis. Scale bar: 100 µm. (**C**) Mouse skin was cultured ex vivo for 24 h in the absence (Ctr) or presence of 50 µM or 100 µM iron (Fe). *S100A8* and *S100A9* gene expression in the epidermis was analyzed by RT-qPCR. (**D**) Bone marrow cells from WT mice were differentiated to wound-healing associated macrophages (Ma(IL4)) and stimulated with 50 µM and 100 µM iron (Fe) for 24 h. *TNF* and *IL-1β* gene expression was analyzed by RT-qPCR. (**E**) Relative gene expression of *TNF* and *IL-1β* was detected by quantitative RT-qPCR in normal skin (ns) and wounds (d7) of control (Ctr) and iron-overloaded (iron) mice. Each dot represents one wound. Mann–Whitney test: * *p* < 0.05, ** *p* < 0.01, *** *p* < 0.001, **** *p* < 0.0001.

**Figure 4 ijms-25-09980-f004:**
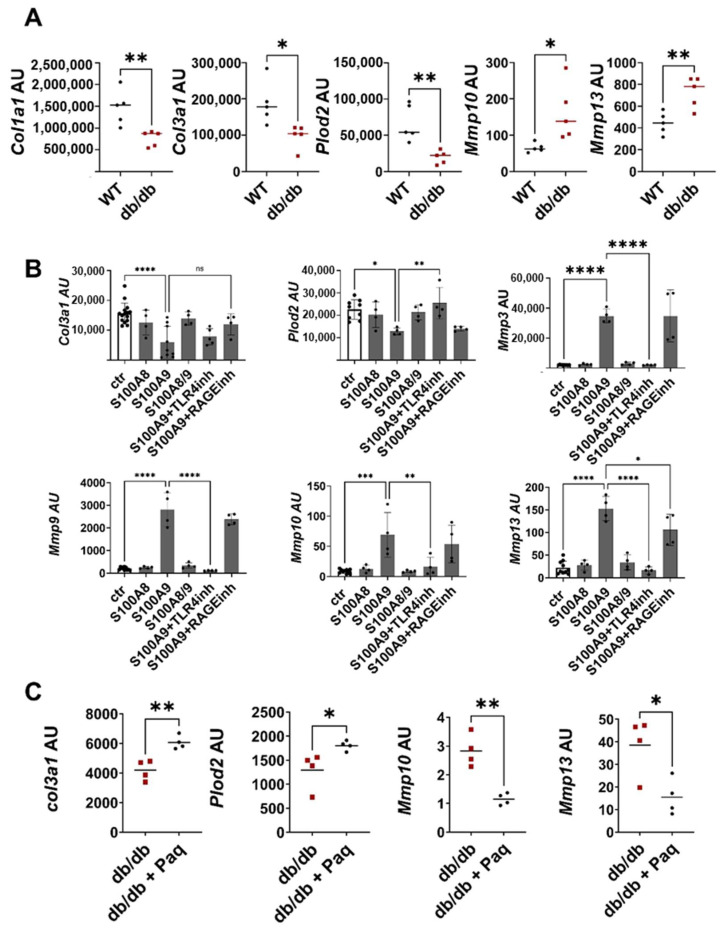
S100A9 directly impairs deposition of ECM by dermal fibroblasts (**A**) Wild-type (WT) and db/db mice were wounded with 6mm punch biopsies. Fibroblasts were isolated 10 days after wounding. Relative expression of indicated genes detected by RT-qPCR in fibroblasts isolated from the wounds of WT and db/db mice. (**B**) Dermal fibroblasts were stimulated in vitro with S100A8, S100A9 and S100A8/9 in the presence or absence of TLR4 (TLR4inh) or RAGE (RAGEinh) inhibitors for 24 h. Relative expression of indicated genes was detected by RT-qPCR. (**C**) Db/db mice were wounded with 6mm punch biopsies. Paquinimod (Paq) was applied 2 days after wounding (db/db + Paq). Fibroblasts were isolated 10 days after wounding. Relative expression of indicated genes in isolated fibroblasts detected by quantitative RT-PCR. Each dot represents one mouse. (**A**,**B**) Mean or mean +/− standard deviation (**C**) is indicated. (**A**,**C**) Unpaired *t*-test. (**B**) ANOVA with multiple comparisons. ns, not significant, * *p* < 0.05, ** *p* < 0.01, *** *p* < 0.001, **** *p* < 0.0001.

## Data Availability

The data that support the findings of this study are available from the corresponding author upon reasonable request.

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
