# Peer review of "Dysregulated S100A9 Expression Impairs Matrix Deposition in Chronic Wounds"

_ijms, 2024, doi:10.3390/ijms25189980_

Round 1

Reviewer 1 Report

Comments and Suggestions for Authors

1. The Abstract section should be enhanced: key results and central conclusions of the paper need to be emphasized using clearer and more precise language.

2. What are the principal obstacles in transitioning S100A9 from laboratory settings to practical applications? Including a brief commentary on challenges and future directions within the manuscript would be beneficial.

3. Enhance the background descriptions of wound healing by referencing 10.1002/adfm.202400489 and 10.1016/j.carbpol.2024.122064. Additionally, how does the current research improve upon previously published studies in similar areas?

4. The article contains formatting inconsistencies. Please review the usage of dashes and hyphens. Ensure abbreviations are used correctly and consistently throughout the document.

5. The manuscript should include a Figure Abstract to provide a concise overview, thereby enhancing the focus and online visibility of the article. This enables readers to swiftly grasp the main points of the content in a more efficient and clear manner.

Author Response

  1. The Abstract section should be enhanced: key results and central conclusions of the paper need to be emphasized using clearer and more precise language.

As suggested by the reviewer the abstract has been revised.

  1. What are the principal obstacles in transitioning S100A9 from laboratory settings to practical applications? Including a brief commentary on challenges and future directions within the manuscript would be beneficial.

We included the following paragraph at the end of the discussion sections (line 502-524) where we discuss these important aspects:

In our study, we described a local overexpression and uncontrolled activity of S100A9 contributing to disturbed healing of chronic wounds. Consequently, local inhibition of S100A9 activity in the skin wounds would be a favorable approach for clinical transition. Paquinimod, Tasquinimod and ABR-238901 are potent S100A9 inhibitors through blocking the interaction of S100A8/A9 with its receptors RAGE (receptor for advanced glycation endproducts) and TLR4 (toll-like receptor 4). Treatments with the S100A8/A9 blockers have demonstrated encouraging results in experimental and clinical interventional studies on cardiovascular diseases, systemic inflammatory and autoimmune diseases and cancer. For example, paquinimod has been effectively tested in phase II clinical trials for the treatment of systemic sclerosis. Therefore, paquinimod may also represent a promising treatment option for chronic wounds with increased S100A9 activity as we showed in our diabetic wound healing model in the mouse. However, Paquinimod, Tasquinimod and ABR-238901 are orally active substances and act systemically bringing the risk of unwanted systemic side effects. One important challenge in the use of S100A8/A9 as a therapeutic target in chronic wounds is the relative abundance of this protein in human circulation, with median levels of approximately 5 mg/L in healthy individuals, rising up to 15 mg/L in atherosclerosis patients. From this perspective, the development of topical application methods for S100A9 inhibitors for local application to the skin or wounds would be an important approach for the clinical implementation of our findings.

  1. Enhance the background descriptions of wound healing by referencing 10.1002/adfm.202400489 and 10.1016/j.carbpol.2024.122064. Additionally, how does the current research improve upon previously published studies in similar areas?

We thank the Reviewer for this note. We included in the manuscript (line 42) the reference to the most recent review on cellular and molecular mechanism of skin wound healing published this year in Nature Reviews Molecular Cell Biology.

Previously published studies, including ours [10.7150/thno.67174; 10.1016/j.bioactmat.2021.04.026; 10.1126/scitranslmed.aai9044.], addressed the role of excessive inflammation and its modulation in disturbed healing chronic wounds, primarily focusing on the effects on immune cells, particularly macrophages. In this study, we describe a mechanism for how the increased activity of an inflammatory factor, S100A9, directly compromises the functions of resident skin cells and fibroblasts and how blocking of this factor can restore the function of fibroblasts. Fibroblasts are increasingly recognized as key regulators of inflammation and repair processes, and understanding their dysregulation in impaired wound healing potentially opens up new targets for therapies. We emphasized these aspects in the manuscript line 427-430 and line 498-504

  1. The article contains formatting inconsistencies. Please review the usage of dashes and hyphens. Ensure abbreviations are used correctly and consistently throughout the document.

We have revised the manuscript and corrected the usage of dashes and hyphens and abbreviations in the text, figures and figures legend.

  1. The manuscript should include a Figure Abstract to provide a concise overview, thereby enhancing the focus and online visibility of the article. This enables readers to swiftly grasp the main points of the content in a more efficient and clear manner.

Thanks for the suggestion. We included a Graphical Abstract summarizing our findings.

Reviewer 2 Report

Comments and Suggestions for Authors

The manuscript with ID: ijms-3193288 titled “Dysregulated S100A9 expression impairs matrix deposition in chronic wounds” is a scientific research where the authors demosntrated the positive action of S100A9 in chronic wounds favouring the activation and deposition of fibroblast in the extracellular matrix junctions using mice models. These findings could be extrapolated to human bionanomedicine approaches. However, some aspects are required to be addressed before to consider this work for further publication in the International Journal of Molecular Sciences.

1) Keywords. The authors should consider to add the term “S100A8” in the keyword list.

2) “In a previous study, (…) dysregulation of S100A8 and A9 (…) induced skin inflammation in obese mice (…) overexpression of S100A8 (A9 during inflammatory conditions in the skin in obesity impairs appropriate macrophage activation (…) inflammation-associated diseases” (lines 76-93). Here, even if I agree with this statement provided by the authors it needs to be also mentioned how S100A9 protein is highly regulated by the environmental calcium concentration [1] limiting the conformational plasticity and the formation of protein fibrils [2] which could lead to neurodegenerative diseases.

[1] Sanders, E.; et al. The Stabilization of S100A9 Structure by Calcium Inhibits the Formation of Amyloid Fibrils. Int. J. Mol. Sci. 2023, 24, 13200. https://doi.org/10.3390/ijms241713200

[2] Carapeto, A.P.; et al. Morphological and Biophysical Study of S100A9 Protein Fibrils by Atomic Force Microscopy Imaging and Nanomechanical Analysis. Biomolecules 2024, 14, 1091. https://doi.org/10.3390/biom14091091

3) Figure 1, panel c (line 122). The lateral scale bar should be added to the respective staining tissue images. Same comment for the Fig. 2, panel c (line 165) and the Fig. 3, panel B (line 221).

4) Did the authors experience any issues related to the normalization of genome-wide expression analysis measurements which could negatively affect to the data interpretation? Some information should be furnished in this regard.

5) Discussion (lines 298-450). This section perfectly remarks the most relevant outcomes found by the authors in this work. No actions are requested from the authors.

6) Materials & Methods (lines 451-519). All the software tools used in this research to process the raw data should be stated in this section.

Author Response

The manuscript with ID: ijms-3193288 titled “Dysregulated S100A9 expression impairs matrix deposition in chronic wounds” is a scientific research where the authors demosntrated the positive action of S100A9 in chronic wounds favouring the activation and deposition of fibroblast in the extracellular matrix junctions using mice models. These findings could be extrapolated to human bionanomedicine approaches. However, some aspects are required to be addressed before to consider this work for further publication in the International Journal of Molecular Sciences.

1) Keywords. The authors should consider to add the term “S100A8” in the keyword list.

We included S100A8 in the key word list.

2) “In a previous study, (…) dysregulation of S100A8 and A9 (…) induced skin inflammation in obese mice (…) overexpression of S100A8 (A9 during inflammatory conditions in the skin in obesity impairs appropriate macrophage activation (…) inflammation-associated diseases” (lines 76-93). Here, even if I agree with this statement provided by the authors it needs to be also mentioned how S100A9 protein is highly regulated by the environmental calcium concentration [1] limiting the conformational plasticity and the formation of protein fibrils [2] which could lead to neurodegenerative diseases.

[1] Sanders, E.; et al. The Stabilization of S100A9 Structure by Calcium Inhibits the Formation of Amyloid Fibrils. Int. J. Mol. Sci. 2023, 24, 13200. https://doi.org/10.3390/ijms241713200

[2] Carapeto, A.P.; et al. Morphological and Biophysical Study of S100A9 Protein Fibrils by Atomic Force Microscopy Imaging and Nanomechanical Analysis. Biomolecules 2024, 14, 1091. https://doi.org/10.3390/biom14091091

We thank the reviewer for this comment. We included information about the relation of environmental calcium concentration and S100A9 function in the Discussion (line 466-471) in the revised manuscript

3) Figure 1, panel c (line 122). The lateral scale bar should be added to the respective staining tissue images. Same comment for the Fig. 2, panel c (line 165) and the Fig. 3, panel B (line 221).

We included bigger scale bars in Fig.1C, Fig.2C and Fig.3B. Length of scale bars is included in the figures legend.

4) Did the authors experience any issues related to the normalization of genome-wide expression analysis measurements which could negatively affect to the data interpretation? Some information should be furnished in this regard.

We do not experience any issues. As described earlier, for data analysis, Affymetrix Gene Chip data were extracted from fluorescence intensities and were scaled in order to normalize data for inter-array comparison using Transcriptome Analysis Console (TAC) 4.0.2 software according to manufacturer’s instruction (Thermo Fisher Scientific). We included the statement in Material and Methods section. Data validation by PCR supports the genome wide expression data.

5) Discussion (lines 298-450). This section perfectly remarks the most relevant outcomes found by the authors in this work. No actions are requested from the authors.

6) Materials & Methods (lines 451-519). All the software tools used in this research to process the raw data should be stated in this section

We included this information in the Materials & Methods section. In particular, quantitative real-time PCR was performed on real-time Thermocycler using qPCR-soft software with Luna Universal qPCR Mastermix (NEB) according to the manufacturer′s instructions.

Affymetrix Gene Chip data were scaled in order to normalize data for inter-array comparison using Transcriptome Analysis Console (TAC) 4.0.2 software according to manufacturer’s instruction. Additionally, we used KEYENCE BZ-9000 fluorescence microscope for capturing images of tissue section stainings and processed the figures with system software from KEYENCE.

Reviewer 3 Report

Comments and Suggestions for Authors

By affecting ECM deposition in fibroblasts, the authors used their study to further demonstrate the pathological function of S100A9 overexpression in injury healing.S100A9 acts directly on fibroblasts through TLR-4 signalling, altering the balance of expression of ECM proteins and ECM-degrading enzymes, leading to a reduction in ECM deposition and subsequently impeding wound healing. Importantly, blocking S100A9 restored fibroblast ECM function and reversed the negative effects on wound healing. Thus, inhibition of S100A9 may be a promising therapeutic approach to improve diabetic wound healing and disturbed tissue repair in chronic venous leg ulcers. Needless to say, this is an interesting attempt, but minor modifications are still required before it can be considered for publication.

1. A detailed methodology on how to model systemic iron overload should be added to sentence 111 of the Results section of this article, or later in the Materials and Methods section.

2. It is necessary for the authors to add an elaboration of the role or mechanism that S100A8/A9 has been shown to play in response to infection, injury, or inflammation prior to this study in Section 1, lines 82 and 83. If necessary, create pictures of the mechanism.

3.It is necessary to include pictures of the control group for comparison in picture 1 for C and G. This will allow for a more visual comparison of wound healing in each condition.

4. The authors need to clarify the logical relationship between sentences 402 and 403 in the Discussion section in more detail.

5. The authors did not provide enough information in the Discussion section about the potential for practical application of the study and should have added the limitations that still exist in this study.

6. There is a need for the authors to improve the writing of the manuscript. This will help to improve the readability of this manuscript.

Comments on the Quality of English Language

Minor editing of English language required.

Author Response

By affecting ECM deposition in fibroblasts, the authors used their study to further demonstrate the pathological function of S100A9 overexpression in injury healing.S100A9 acts directly on fibroblasts through TLR-4 signalling, altering the balance of expression of ECM proteins and ECM-degrading enzymes, leading to a reduction in ECM deposition and subsequently impeding wound healing. Importantly, blocking S100A9 restored fibroblast ECM function and reversed the negative effects on wound healing. Thus, inhibition of S100A9 may be a promising therapeutic approach to improve diabetic wound healing and disturbed tissue repair in chronic venous leg ulcers. Needless to say, this is an interesting attempt, but minor modifications are still required before it can be considered for publication.

  1. A detailed methodology on how to model systemic iron overload should be added to sentence 111 of the Results section of this article, or later in the Materials and Methods section.

In the systemic iron overload model, mice are subjected to repeated treatment with iron-dextran, which results in abundant accumulation of iron in the skin. We added this information in the manuscript (line 135-137).

  1. It is necessary for the authors to add an elaboration of the role or mechanism that S100A8/A9 has been shown to play in response to infection, injury, or inflammation prior to this study in Section 1, lines 82 and 83. If necessary, create pictures of the mechanism.

We included more information in the revised manuscript in line 84-102.

  1. It is necessary to include pictures of the control group for comparison in picture 1 for C and G. This will allow for a more visual comparison of wound healing in each condition.

Figure 1C and D represent the respective control wounds to the wounds of our models: Wildtype (wt) in Fig.1C is the control for the diabetic db/db model. Ctr in Fig.1D is the untreated (no iron) control for the systemic iron model.

  1. The authors need to clarify the logical relationship between sentences 402 and 403 in the Discussion section in more detail.

Following the reviewer suggestion, we have clarified the statement.

  1. The authors did not provide enough information in the Discussion section about the potential for practical application of the study and should have added the limitations that still exist in this study.

We included the following paragraph at the end of the Discussion section (line 505-524) where we discussed these aspects:

In our study, we described a local overexpression and uncontrolled activity of S100A9 contributing to disturbed healing of chronic wounds. Consequently, local inhibition of S100A9 activity in the skin wounds would be a favorable approach for clinical transition. Paquinimod, Tasquinimod and ABR-238901 are potent S100A9 inhibitors through blocking the interaction of S100A8/A9 with its receptors RAGE (receptor for advanced glycation endproducts) and TLR4 (toll-like receptor 4). Treatments with the S100A8/A9 blockers have demonstrated encouraging results in experimental and clinical interventional studies on cardiovascular diseases, systemic inflammatory and autoimmune diseases and cancer. For example, paquinimod has been effectively tested in phase II clinical trials for the treatment of systemic sclerosis. Therefore, paquinimod may also represent a promising treatment option for chronic wounds with increased S100A9 activity as we showed in our diabetic wound healing model in the mouse. However, Paquinimod, Tasquinimod and ABR-238901 are orally active substances and act systemically bringing the risk of unwanted systemic side effects. One important challenge in the use of S100A8/A9 as a therapeutic target in chronic wounds is the relative abundance of this protein in human circulation, with median levels of approximately 5 mg/L in healthy individuals, rising up to 15 mg/L in atherosclerosis patients. From this perspective, the development of topical application methods for S100A9 inhibitors for local application to the skin or wounds would be an important approach for the clinical implementation of our findings.

  1. There is a need for the authors to improve the writing of the manuscript. This will help to improve the readability of this manuscript.

The manuscript has been edited by an English expert.